# Improved Time-Synchronization Algorithm Based on Direct Compensation of Disturbance Effects

**DOI:** 10.3390/s19163499

**Published:** 2019-08-10

**Authors:** Yeoung-Duk Seo, Kyou Jung Son, Gi-Sung An, Kyung Deok Nam, Tae Gyu Chang, Sang-Hee Kang

**Affiliations:** 1School of Electrical and Electronics Engineering, Chung-Ang University, Seoul 06974, Korea; 2Department of Electrical Engineering, Myongji University, Yongin 17058, Korea

**Keywords:** IEEE 1588 Precision Time Protocol (PTP), Ethernet-based industry-plant monitoring and control, precision time synchronization, stochastic model-based estimator, IEC 61850-based digital substation

## Abstract

In this paper, an improved time-synchronization algorithm is proposed. The improvement of time synchronizing performance was achieved by introducing a stochastic model-based direct compensation of the disturbance effects appearing in the IEEE 1588 Precision Time Protocol (PTP)-based time synchronization system. A dynamic model of PTP clock system was obtained by reflecting the three major sources of disturbances, i.e., clock frequency drift, clock rate offset, and network noise. With the application of the dynamic model of the PTP clock system, the effects of the disturbances can be effectively eliminated in the PTP time synchronization control loop. Computer simulations are performed to verify the performance of the proposed time synchronization algorithm by applying the various types of disturbances, including network noise and clock drift. The simulation results are compared with those of other representative time synchronization algorithms, i.e., IEEE 1588 PTP algorithm and Kalman-filter-based algorithm. It is shown that the proposed algorithm improves time synchronizing performance up to 84% with respect to that of the Kalman-filter-based synchronization algorithm when simulated with colored noise type disturbances. The proposed time synchronization algorithm is expected to contribute for the realization of future Ethernet-based industry-plant monitoring and control including IEC 61850-based digital substation.

## 1. Introduction

Ethernet-based provision of precise time information has been an important issue for its industry application where the required time precision is very high. The IEEE 1588 Precision Time Protocol (PTP) [1] was published as a standard for the provision of Ethernet-based precise time-synchronization means for such industry applications. The IEEE 1588 PTP-based time synchronization mechanisms, as shown by many previous studies [2,3,4,5], are based on the basic functional elements to estimate a local clock’s time offset in reference to a master clock and to adjust the local clock’s time and frequency. IEEE standard c37.238-2011, 2017 [6,7] was published to provide required specifications, including less than 1.0 microsecond of timing error bound, for advanced operation and protection of IEC 61850-based digital substations [8,9]. The standard for communication networks and systems for power utility automation added the precision time protocol profile, i.e., IEC 61850-9-3 in 2016 [10], for compliant use of the communication protocol in digital substations.

Wide variability and disturbances, which are considered as inherent characteristics of an Ethernet-based network environment, are two of the technical difficulties preventing the wide spread of practical PTP applications in power systems [11,12,13,14,15,16,17]. There have been many efforts to provide an adaptive method to compensate the effects of variability and disturbances of network environment [18,19,20], eventually to achieve an effective and fault-tolerant control of time synchronization [21]. Proper modeling of a disturbance environment and its reflection into a PTP time synchronization control algorithm should be emphasized for effective and satisfactory compensation of the effects of disturbances [18].

This paper proposes an improved time synchronization control algorithm which adaptively reduces the effect of disturbances to minimize its performance degradation. A dynamic model of a PTP clock system was obtained by reflecting the three typical sources of disturbances, i.e., clock frequency drift, clock rate offset, and network noise, which are the major components consisting the overall disturbances. Base on the dynamic model, a direct compensation algorithm is proposed where the effect of the disturbances is estimated by stochastic modeling and is subtracted from the PTP time synchronization control loop. Arbitrary patterns of disturbances can be effectively estimated by autoregression (AR)-based stochastic modeling. The proposed PTP time synchronization control algorithm is considered to provide an effective joint handling of disturbance estimation and the dynamic model of PTP clock, yielding minimized performance degradation.

Based on the dynamic model, which effectively reflects the three typical sources of disturbances, the PTP time synchronization control system’s stability bound was analytically derived as a closed form inequality function. The stability bound shows the allowable range of PTP syntonization gain, i.e., clock frequency control gain and time synchronization interval, and provides a basis for the optimized design of the adaptive PTP time synchronization control algorithm. The optimization based on the dynamics modeling of PTP clock systems provides the algorithmic solution for the inadequate limits of previous research. The proposed time synchronization algorithm is expected to contribute to the realization of a future Ethernet-based industry-plant monitoring and control.

Computer simulations were conducted to demonstrate the performance of the proposed algorithm. The performance of the proposed PTP control algorithm was obtained by simulating the algorithm under various disturbance environments. From the simulation results, it was confirmed that the time synchronization error caused by the disturbances was effectively reduced. To verify the excellence in precise time synchronization of the proposed algorithm, time synchronizing performances of the proposed algorithm were compared with the IEEE 1588 PTP algorithm [1] and Kalman-filter-based algorithm [18]. It was confirmed that the proposed algorithm relatively reduces the root mean square (RMS) value of time offset up to 4.5% and 3% for the white-noise type disturbances with respect to the IEEE 1588 PTP algorithm and Kalman-filter-based algorithm, respectively. For the colored-noise type disturbances, the improvement was achieved up to 84% with respect to the other algorithms.

This paper is organized as follows. In Section 2, the dynamic model of PTP clock systems reflecting the effects of typical disturbances is presented and the adaptive time synchronization control algorithm is proposed. The performances of the proposed algorithm are evaluated with computer simulations in Section 3. Finally, the conclusions are given in Section 4.

## 2. Adaptive PTP Time Synchronization Control Algorithm

The effects of typical disturbances on the PTP time synchronization control system studied and the dynamic model obtained by reflecting the effects of disturbances are explained in Section 2.1. Based on the dynamic model, a PTP time synchronization control system’s stability bound is analytically derived as a closed form inequality function in Section 2.2. In Section 2.3, an adaptive PTP time synchronization control algorithm is proposed to effectively suppress the effects of the three typical disturbances. The proposed adaptive PTP algorithm is based on a direct compensation of the effects of the disturbances from the time offset signal in the PTP time synchronization control loop. An AR model-based stochastic approach was adopted for the optimized estimation of the spectral characteristic of the disturbances.

### 2.1. Analysis of the Effects of Disturbances on PTP Time Synchronization

In this subsection, a dynamic model is obtained by reflecting the three typical sources of disturbances, i.e., clock frequency drift, clock rate offset, and network noise, which are the major components consisting the overall disturbances in PTP time synchronization system. The analysis provides an analytical tool for a derivation of stability bound in Section 2.2 and a design of optimized control in Section 2.3.

The functional block diagram suggested for the dynamics modeling of the PTP time synchronization control system is presented in Figure 1. The local PTP clock system can be divided into two blocks, i.e., the local clock and the PTP control block, respectively. The local clock updates the device’s present time, which is marked as Time, using a hardware-based clock oscillator, which is marked as Oscillator. The PTP control block consists of two major functions, i.e., time synchronization and the frequency syntonization. The time synchronization updates the present time of the local clock and computes the offset between the reference clock time and the local clock time. The computed offset is used in the frequency syntonization block for the adjustment of the clock rate of the local clock’s oscillator. The major error-causing disturbances considered in the dynamics modeling are the clock frequency drift, the network disturbance, and the clock frequency offset.

In the following, a detailed description of the dynamic model of PTP time synchronization control system is shown with Equations (1)–(3), which correspond to the time synchronization block and the frequency syntonization block:(1)$$y\left(n\right)=x_{1}\left(n\right)+T_{S}x_{2}\left(n\right)-\overset{´}{r}\left(n\right)$$
(2)$$x_{1}\left(n+1\right)=x_{1}\left(n\right)+T_{S}x_{2}\left(n\right)-y\left(n\right)$$
(3)$$x_{2}\left(n+1\right)=x_{2}\left(n\right)-ky\left(n\right)$$
where the absolute value of time and frequency of the local clock are defined as the two-state variables $x_{1}\left(n\right)$ and $x_{2}\left(n\right)$, respectively. The measured reference clock time, which is provided with the reference clock at every sync interval $T_{S}$, is reflected as an independent input $\overset{´}{r}\left(n\right)$.

As shown in Equation (1), the measured time offset y(n) is obtained by subtracting the measured reference clock time $\overset{´}{r}\left(n\right)$ from the updated local clock time $x_{1}\left(n\right)+T_{S}x_{2}\left(n\right)$. The time synchronization mechanism of the PTP is shown with Equation (2), where the time synchronization is performed by subtracting the measured time offset y(n) from the updated local clock time. The frequency syntonization mechanism of the PTP clock rate control is represented in Equation (3), where the updated local clock rate $x_{2}\left(n+1\right)$ is obtained by applying the negative feedback ky(n) to the previous clock rate $x_{2}\left(n\right)$. It is an advantageous feature of the proposed model represented with Equations (1)–(3) that a more detailed description and PTP time synchronization control can be achieved in comparison with other previous models [18], where the two-state variables correspond to the relative offset and rate difference between the master and the local clock, respectively. The advantage comes from the increased number of description features of the clock behaviors in the proposed model, where the absolute values of time and frequency of local clock are defined as the two-state variables and those of the master clock are reflected as an independent input and as an initial value of state variable, respectively.

The effects of the disturbances are reflected in Equation (4):(4)$$\overset{´}{r}\left(n\right)=r\left(n\right)+\upsilon \left(n\right)$$
where r(n): the reference clock time and υ(n): the network disturbance. The effect of network disturbance is reflected in the process of obtaining the measured reference clock time $\overset{´}{r}\left(n\right)$, as shown in Equation (4), where the measured reference clock time is obtained by adding the network disturbance υ(n) to the reference clock time r(n).
(5)$$x_{2}\left(n\right)=\frac{f_{0}+f_{off}\left(n\right)+\overset{´}{d}\left(n\right)}{f_{0}}=\frac{f_{0}+f_{off}\left(n\right)}{f_{0}}+\frac{\overset{´}{d}\left(n\right)}{f_{0}}={\tilde{x}}_{2}\left(n\right)+d\left(n\right)$$
where $f_{0}$: the nominal local clock frequency, $f_{off}\left(n\right)$: the frequency offset between the reference clock and the local clock, $\overset{´}{d}\left(n\right)$: the local clock frequency drift, ${\tilde{x}}_{2}\left(n\right)$: the normalized local clock rate, and d(n): the normalized local clock frequency drift. The clock rate is modeled as a superposition of the nominal frequency, offset, and the clock frequency drift as explained with the proposed dynamic model of Figure 1. As shown in Equation (5), the normalized values, ${\tilde{x}}_{2}\left(n\right)$ and d(n), are used as the clock rate offset and the clock frequency drift, respectively.

### 2.2. Stability Analysis Based on Dynamics Modeling of PTP Time Synchronization System

In this subsection, the PTP time synchronization control system reflecting the effects of disturbances in Section 2.1 is modeled, and its stability bound is analytically derived. The analyses were based on the state-equations of the PTP time synchronization control system. The PTP clock system’s stability bound was derived as a closed form inequality function in terms of the two dominant system parameters, i.e., the synchronization interval, which is a time interval between two IEEE 1588 sync messages, and the syntonization feedback gain. The dynamic model of the PTP time synchronization control system and its stability bound provides a basis for the optimized design of the adaptive PTP control algorithm presented in Section 2.3.

The signal flow diagram of the dynamic models represented with Equations (1)–(5) is shown in Figure 2.

The state equation representation of the signal flow diagram is shown in Equations (6) and (7):(6)$$x\left(n+1\right)=Ax\left(n\right)+(K_{1}+K_{2})\left\{Cx\left(n\right)-\overset{´}{r}\left(n\right)\right\}$$
(7)$$y\left(n\right)=Cx\left(n\right)-\overset{´}{r}\left(n\right)$$
where $A=\;\left(\begin{array}{cc} 1 & T_{s}\\ 0 & 1 \end{array}\right)$, $K_{1}=\left(\begin{array}{c} -1\\ 0 \end{array}\right)$, $x\left(n\right)=\left(\begin{array}{c} x_{1}\left(n\right)\\ x_{2}\left(n\right) \end{array}\right)$, $C=\left(1\;T_{s}\right)$, $K_{2}=\left(\begin{array}{c} 0\\ -k \end{array}\right)$.

The stability bound of the PTP time synchronization can be straightforwardly obtained by finding the system’s pole locations with its state Equations (7) and (8). The stability bound is derived as a closed form functional representation in terms of the two key performance determining factors $T_{s}$, k, which are the syntonization gain and the synchronization interval, respectively.

The pole locations are obtained from the system’s characteristic polynomial shown in Equation (8):(8)$$\det\left(\begin{array}{cc} -\lambda & 0\\ -k & 1-kT_{s}-\lambda \end{array}\right)=\;\lambda^{2}+\left(kT_{s}-1\right)\lambda =0$$
then: $p_{1}=1-kT_{s}$, $p_{2}=0$.

From Equation (8), it can be known that one pole is located at $1-kT_{s}$ and the other pole is located at the origin. The stability bound of the syntonization gain k is represented in Equation (9):(9)$$0<k<\;\frac{2}{T_{s}}.$$

As shown in Equation (9), the stability bound is determined by the syntonization gain k and the synchronization interval $T_{s}$.

The stability bound curve of k with respect to $T_{s}$ is shown in Figure 3, where a stable region is represented as below the curve. As $T_{s}$ is determined, the value of k must be chosen not to be located over the curve. Equation (9) implies that a shorter synchronization interval results in a wider selectable range of syntonization gain.

### 2.3. Adaptive PTP Time Synchronization Algorithm

In this subsection, an adaptive PTP time synchronization control algorithm is proposed to effectively suppress the effects of the three typical disturbances reflected in the PTP clock model obtained in Section 2.1. The proposed adaptive algorithm is based on a direct compensation of the effect of the disturbances from the time offset in the PTP time synchronization control loop. An AR model-based stochastic approach was adopted for the optimized estimation of the spectral characteristic of the disturbances, i.e., slow tonic nature of clock frequency drift perturbed with small jitters.

The functional block diagram of the PTP time synchronization control system, including the proposed adaptive syntonization algorithm, is shown in Figure 4. The proposed PTP time synchronization control system including direct compensation consists of the major functional blocks, i.e., the AR model-based disturbance remover, the recursive least square (RLS)-based adaptive weight-control mechanism, and the clock rate calculator with a feedback controller. The filter coefficients of the disturbance remover are adaptively updated to reduce the disturbance effect from the time offset.

The signal flow diagram of the proposed PTP time synchronization system with direct compensation is shown in Figure 5. The structure of the proposed algorithm is based on the application of linear prediction filter and RLS algorithm to estimate the disturbance and its direct compensation from y(n). The signal ξ(n), which represents disturbance-removed residue, is used as the feedback signal for the clock rate syntonization.

The time synchronizing performance, including the stability margin, is effectively improved by the direct compensation of the disturbance effects. The AR model-based least square solutions, which are based on their second-order statistics, are considered effective in removing the effects of the disturbances. The advantage of the proposed PTP time synchronization control algorithm is also supported from the slowly changing tonic nature of the clock drift, which can be properly modeled with finite number of poles. The update rules of the RLS algorithm are summarized in Table 1.

## 3. Simulations and Results

In this section, the performances of the proposed PTP time synchronization algorithm are evaluated with computer simulations. To verify the excellence of the proposed algorithm, the time synchronizing performances of the proposed algorithm were compared with IEEE 1588 PTP [1] and Kalman-filter-based clock synchronization algorithm [18]. Network noise and clock drift were applied to time synchronizing environments as disturbances. 

For the application of disturbances to time synchronizing environments, the clock drift was modeled as a sinusoidal waveform where the frequency is 0.025 Hz, and the network noise was modeled as an additive white gaussian noise and colored noise. The amount of the effect of clock drift was controlled by varying the RMS value of the clock drift, which is represented as Equation (10):(10)$$d_{RMS}=\sqrt{\frac{1}{N}\overset{N}{\underset{n=1}{\sum }}d{\left(n\right)}^{2}},$$
where $d_{RMS}$: the RMS value of clock drift waveform, N: the total number of the time synchronization process, and d(n): the values of clock drift waveform at the instant n. For the comparison of the time synchronizing performances, the RMS values of time offset were obtained for the three algorithms using Equation (11):(11)$$t_{off,RMS}=\sqrt{\frac{1}{N}\overset{N}{\underset{n=1}{\sum }}t_{off}{\left(n\right)}^{2}},$$
where $t_{off,RMS}$: the RMS value of time offset data, N: the total number of the time offset data, and $t_{off}\left(n\right)$: the time offset measured at the instant n.

The improved time synchronizing performance of the proposed algorithm is illustrated with the examples of simulation results in Figure 6. The disturbance affected time offset is significantly reduced by applying the proposed algorithm, as shown in Figure 6a. By comparing the power spectrums of the measured time offset and that of the estimated time offset as shown in Figure 6b,c, respectively, it is also observed that the disturbance affected time offset is effectively estimated by the stochastic model-based disturbance estimator. In the simulation, the synchronization interval is set to 4 s, the initial clock rate of the reference clock and local clock are set to 100 Hz and 150 Hz, respectively, and the length of the RLS filter is set to 10. The RMS value of the clock drift is set to $1.5\times 10^{-2}$ Hz, and the variance of network noise is set to $2.5\times 10^{-5}$
$s^{2}$. The power spectrum of the time offset signal is obtained by Equation (12):(12)$$Y_{\mathrm{PSD}}\left[\omega \right]={\left\{\frac{1}{N}\overset{N}{\underset{n=1}{\sum }}t_{off}\left(n\right)e^{-j\omega n}\right\}}^{2},$$
where $Y_{\mathrm{PSD}}\left[\omega \right]$: the power spectrum of the time offset data and N: the total number of the time offset data.

The simulation results of the proposed algorithm are summarized in Table 2. The performances of time offset reduction were obtained by applying the disturbances with varying amounts. In the simulations, the variance of the network noise varies from $2.5\times 10^{-7}$ to $3.125\times 10^{-5}$
$s^{2}$, and the RMS value of the clock drift is fixed to 0.01 Hz. In a similar way, the RMS value of the clock drift varies from $3.5\times 10^{-3}$ to $1.5\times 10^{-2}$
Hz, and the variance of the network noise is fixed to $2.5\times 10^{-7}$
$s^{2}$. 

The advantages of the application of the proposed algorithm are also shown by comparing its performance with those of the other representative algorithms, i.e., the IEEE 1588 PTP algorithm and the Kalman-filter-based time synchronization algorithm, under various noise environments. Two types of noise, i.e., white noise and colored noise, were used to for the simulation. The colored noise was modeled as a summation of the two components. One is the additive white gaussian noise and the other one is the sinusoidal waveform which reflects slowly varying characteristic of clock drift. The length of the RLS filter is set to 25 in order to sufficiently reflect the dynamic variations of clock drift.

As shown in Table 3, the proposed algorithm shows better performance than those of the other two algorithms in the case of a white noise environment. The proposed algorithm shows up to 4.5% and 3% of relative time offset reductions with respect to those of IEEE 1588 PTP and the Kalman algorithm when the white noise level is $3\times 10^{-2}$ s^2^. For the colored noise, the proposed algorithm shows more distinctive superiority compared to the other algorithms. It is shown that the relative reduction of 84% is achieved for the colored noise having the variance of $3\times 10^{-5}$ s^2^. This high level of superiority is attributed to the ‘colored-noise representation fidelity’ of the stochastic modeling approach.

## 4. Conclusions

In this paper, an improved PTP time synchronization algorithm is proposed. The proposed algorithm effectively estimates and reduces arbitrary patterns of network and clock-drift disturbances by applying an adaptive estimation of disturbance effects and its direct compensation in the PTP clock synchronization system. It was verified through computer simulations that the proposed algorithm shows better performance than those of the other representative algorithms, i.e., IEEE 1588 PTP and Kalman-filter-based algorithm. Simulation results showed that the proposed algorithm reduces the RMS value of time offset up to 4.5% and 3% with respect to those of IEEE 1588 PTP and the Kalman algorithm when the white noise level is $3\times 10^{-2}$ s^2^. The proposed algorithm, more distinctively, shows 84% of relative reduction for the colored noise with the variance of $3\times 10^{-5}$ s^2^. The proposed time synchronization algorithm is expected to contribute to the realization of a future Ethernet-based industry-plant monitoring and control including IEC 61850-based digital substation.

## Figures and Tables

**Figure 1 sensors-19-03499-f001:**
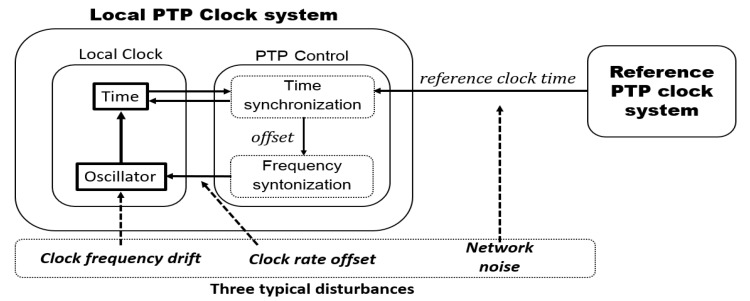
A functional block diagram suggested for the dynamics modeling of the Precision Time Protocol (PTP) clock system reflecting three typical disturbances (clock frequency drift, clock rate offset, and network noise).

**Figure 2 sensors-19-03499-f002:**
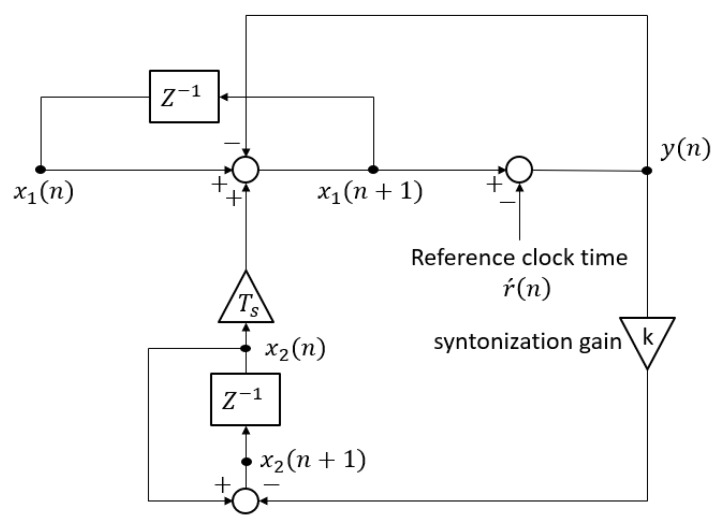
The signal flow diagram of the PTP time synchronization control system obtained from the dynamic model reflecting the disturbances.

**Figure 3 sensors-19-03499-f003:**
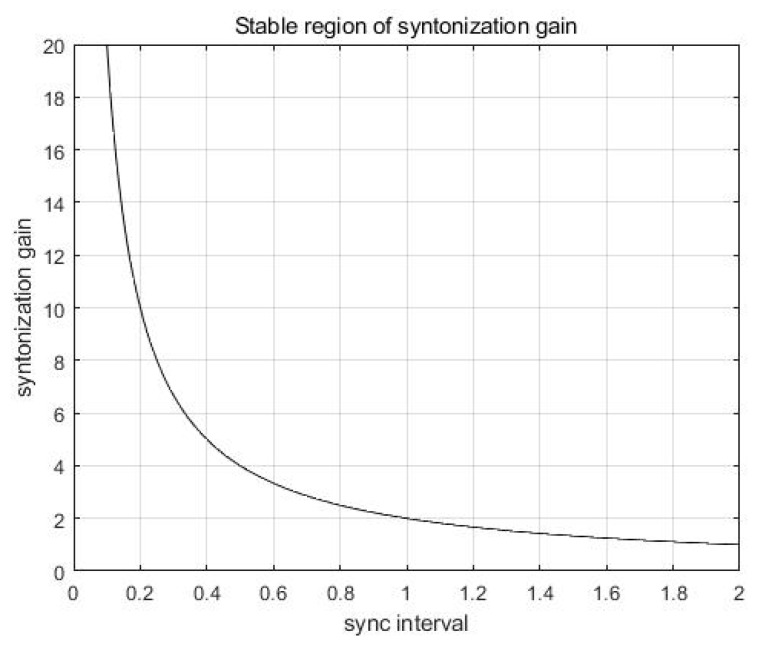
The stable region of syntonization gain.

**Figure 4 sensors-19-03499-f004:**
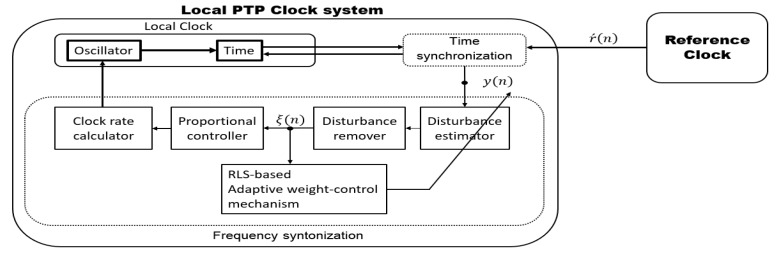
A functional block diagram of the proposed adaptive PTP time synchronization algorithm.

**Figure 5 sensors-19-03499-f005:**
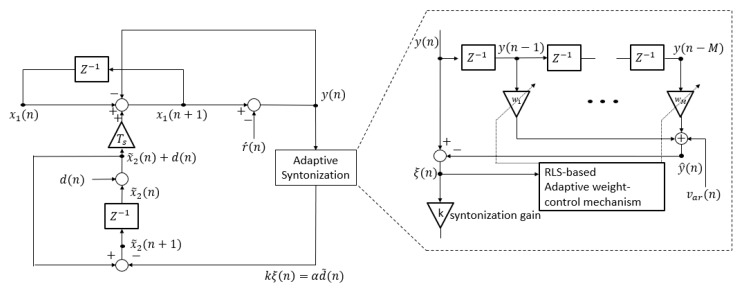
Signal flow diagram of the dynamic modeling of the PTP time synchronization system with direct compensation of the disturbance effects.

**Figure 6 sensors-19-03499-f006:**
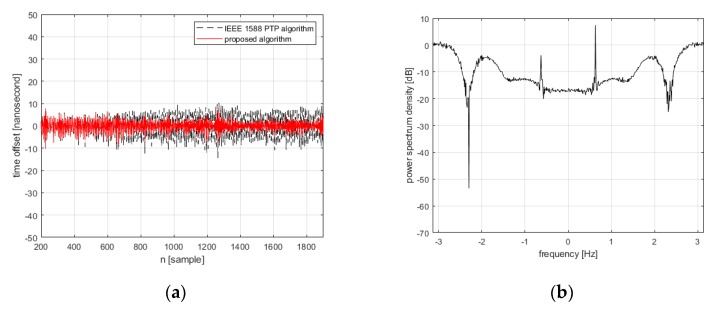
The simulation results of the proposed algorithm with the direct compensation of the disturbance effects. (**a**) time synchronization performances (compensated time offset) of the proposed algorithm, (**b**) power spectrum of the disturbance affected time offset, and (**c**) power spectrum of the estimated time offset.

**Table 1 sensors-19-03499-t001:** The update rules of the recursive least square (RLS) algorithm.

The initial set values of the RLS algorithm is $\bar{w}(0)=0$ $P(0)=\delta^{-1}I$
For each instant of time, n=1,2,…, compute $\pi (n)=P(n-1)\bar{y}(n-1)$, $\bar{k}(n+1)=\frac{\pi (n)}{1+\bar{y}(n-1)\pi (n)}$, $\xi (n)=y(n)-\hat{y}(n)$, $\bar{w}(n+1)=\bar{w}(n)+\bar{k}(n)\xi (n)$, and $P(n)=P(n-1)-k(n)\bar{y}(n-1)P(n-1)$ Note, $\bar{y}(n)=\left(\begin{array}{c} y(n-1)\\ y(n-2)\\ y(n-M) \end{array}\right)$

**Table 2 sensors-19-03499-t002:** Simulation results in various network noises and clock drifts.

Various Disturbances		RMS Values of Time Offset [nanosec]	
		IEEE 1588 PTP	Proposed Algorithm
**Variance of Network Noise [** $s^{2}$ **]**	$2.5\times 10^{-7}$	4.2711	2.0786
	$1.25\times 10^{-6}$	4.3218	2.2311
	$6.25\times 10^{-6}$	4.4134	2.3391
	$3.125\times 10^{-5}$	5.0264	3.0110
**RMS Values of Clock Drift [** Hz **]**	$3.5\times 10^{-3}$	1.3618	1.3856
	$7.0\times 10^{-3}$	2.1465	1.8152
	$1.0\times 10^{-2}$	4.0074	2.3112
	$1.5\times 10^{-2}$	5.8746	2.7911

**Table 3 sensors-19-03499-t003:** Performance summary table of simulation results in various noise environments.

$\mathrm{Variance}\;\mathrm{of}\;\mathrm{Noise}\;[s^{2}]$		Root Mean Square (RMS) Values of Time Offset [s]		
		IEEE 1588 PTP [1]	Kalman Filter Algorithm [18]	Proposed Algorithm
**White Noise**	$3\times 10^{-5}$	0.00964	0.00948	0.00941
	$3\times 10^{-4}$	0.03135	0.03079	0.03035
	$3\times 10^{-3}$	0.10133	0.10013	0.09832
	$3\times 10^{-2}$	0.33051	0.32725	0.31981
**Colored Noise**	$3\times 10^{-5}$	0.05886	0.05928	0.00958
	$3\times 10^{-4}$	0.06561	0.06554	0.02963
	$3\times 10^{-3}$	0.11640	0.11444	0.09564
	$3\times 10^{-2}$	0.31799	0.31360	0.30254

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
