# Peer review of "Improved Time-Synchronization Algorithm Based on Direct Compensation of Disturbance Effects"

_sensors, 2019, doi:10.3390/s19163499_

Round 1

Reviewer 1 Report

The paper is concerned with designing an improved PTP time synchronization control algorithm. The result is correct, while the following comments should be considered. The authors should clarify the model (1)--(3), dose the model give by the authors firstly? The control algorithm with or without adaptive mechanism is simple, and the advantages of the proposed method over the existing ones should be highlighted. The following related result on the synchronization of nonlinear systems should be included to complete the literature review. DOI: 10.1109/TNNLS.2015.2507183  

Reviewer 2 Report

An improved precise time protocol (PTP) time synchronization control algorithm proposed in this study. Comments:

 1. Figure 1 can be adjusted to the correct position.

2. The symbols are not clearly stated and some symbols are not explained.

3. Is the interpretation of Equation (1) wrong? We didn't see Δ?1(?) in the equation.

4. How the equation is derived is not reasonably proved.

5. What does foff(n) stand for in Equation (5)?

6. Mathematical equations without seeing the objective function and the final result of Root Mean Square(RMS) Values of Synchronization Error, of SNR (dB) and of Power Spectrum of Drift Effect.

7. Results should be compared with existing recent techniques.

Author Response

First of all, we thank you very much for your considerations on our submitted manuscript. We revised the manuscript reflecting your all requirements and comments. Please see the attachment.

Reviewer 3 Report

This paper presents an improved PTP time synchronization control algorithm. The authors provide an extensive analysis of the proposed algorithm, while focusing on the trending concept of power systems. However, despite the helpful figures and experimental results the work is not easy to read. Several parts regarding method description tend to be repeated throughout the text, while no explanation is provided regarding the structure of the paper. Furthermore, introduction should include more information regarding the background of PTP. The authors should also try to enhance contribution by providing a comparison between the proposed scheme and other related schemes in the form of a summary table in the according section. Finally, the related work should be thoroughly updated with 2018/2019 papers.

Author Response

(The authors gave the same response as above.)
